# An Automated Light Trap to Monitor Moths (Lepidoptera) Using Computer Vision-Based Tracking and Deep Learning

**DOI:** 10.3390/s21020343

**Published:** 2021-01-06

**Authors:** Kim Bjerge, Jakob Bonde Nielsen, Martin Videbæk Sepstrup, Flemming Helsing-Nielsen, Toke Thomas Høye

**Affiliations:** 1School of Engineering, Aarhus University, Finlandsgade 22, 8200 Aarhus N, Denmark; jakob.bonde.nielsen@gmail.com (J.B.N.); martinvsep@gmail.com (M.V.S.); 2NaturConsult, Skrænten 5, 9520 Skørping, Denmark; naturconsult@hotmail.com; 3Department of Bioscience and Arctic Research Centre, Aarhus University, Grenåvej 14, 8410 Rønde, Denmark; tth@bios.au.dk

**Keywords:** biodiversity, CNN, computer vision, deep learning, insects, light trap, moth, tracking

## Abstract

Insect monitoring methods are typically very time-consuming and involve substantial investment in species identification following manual trapping in the field. Insect traps are often only serviced weekly, resulting in low temporal resolution of the monitoring data, which hampers the ecological interpretation. This paper presents a portable computer vision system capable of attracting and detecting live insects. More specifically, the paper proposes detection and classification of species by recording images of live individuals attracted to a light trap. An Automated Moth Trap (AMT) with multiple light sources and a camera was designed to attract and monitor live insects during twilight and night hours. A computer vision algorithm referred to as Moth Classification and Counting (MCC), based on deep learning analysis of the captured images, tracked and counted the number of insects and identified moth species. Observations over 48 nights resulted in the capture of more than 250,000 images with an average of 5675 images per night. A customized convolutional neural network was trained on 2000 labeled images of live moths represented by eight different classes, achieving a high validation $F_{1}$-score of 0.93. The algorithm measured an average classification and tracking $F_{1}$-score of 0.71 and a tracking detection rate of 0.79. Overall, the proposed computer vision system and algorithm showed promising results as a low-cost solution for non-destructive and automatic monitoring of moths.

## 1. Introduction

There is growing evidence that insect populations are declining in abundance, diversity, and biomass [1]. Multiple anthropogenic stressors are proposed as drivers of these changes [2], but the complexities of the responses and differences in the ecology of the millions of species involved make it difficult to pinpoint causes of these declines as well as mitigation measures [1]. For many taxa, long-term population census data are also non-existing, and more extensive monitoring especially of species-rich taxa is critically needed.

More than half of all described species on Earth are insects. With more than 160,000 described species, the insect order Lepidoptera, which consists of moths (88%) and butterflies (12%), is one of the largest. Moths are important as pollinators, herbivores and prey for, e.g., birds and bats. Reports of declining moth populations have come from Germany [1], Great Britain [3], Sweden [4], and the Netherlands [5]. Recently, however, stable populations have also been observed [6]. Changes in the abundance of moths could have cascading effects through the food web, suggesting that moths are a very relevant group of insects to monitor more effectively in the context of global change of climate and habitats. Some of the most damaging pest species in agriculture and forestry are also moths [7,8] and insects are known to be major factors in the world’s agricultural economy. Therefore, it is crucial to improve insects population monitoring systems.

The most widely applied method of monitoring moths is by using light traps [9]. Moths are typically killed, manually counted, and classified by humans, which requires expert knowledge and is labor-intensive. In this paper, we describe a new method for automatic counting and classification of live moths in order to accelerate the study of moth populations and minimize the impact on their populations.

Some attempts to record moths with computer vision or identify moths based on images have already been made. For instance, Ding and Taylor [10] presented a trap for automatic detection of moths, which contained a pheromone lure to attract insects of interest and an adhesive surface, where the insects became stuck. Digital images with a relatively low resolution (640 × 480 pixels) were taken of the dead and live insects at a fixed time point daily and transmitted to a remote server. The paper showed that it is possible to train a Convolutional Neural Network (CNN) to identify moths. However, it only dealt with a binary classification problem to recognize if a trapped insect was a moth or not and did not consider classification of individual moth species.

A few studies have attempted to classify individual species of moths. Watson et al. [11] and Mayo and Watson [12] have provided an analysis of 774 live individuals from 35 different moth species to determine whether computer vision techniques could be used for automatic species identification. Focusing on data mining for feature extraction and a Support Vector Machine (SVM) for classification, their work achieved a classification accuracy of 85% among 35 classes distributed across 774 images.

While previous analyses of the same dataset would require manual selection of regions on the moths to identify an individual, Mayo and Watson [12] could effectively classify from a single image. In addition, Batista et al. [13] have proposed a nearest neighbor algorithm based on features such as texture, color, and shape to classify moth species from the same dataset, obtaining an accuracy of 79.53%. The work of Watson et al. [11], Mayo and Watson [12], and Batista et al. [13] were published before CNNs became a widespread technology, and their work contributed future work in the area, laying the foundation for solving the problem of species classification with deep learning.

Several of the issues associated with moth classification when the dataset contains a large number of classes have been illustrated by Chang et al. [14]. Their work presented a dataset of 636 butterfly and moth species distributed across 14,270 highly detailed images, which were collected using internet search engines. The challenge with such a large dataset of haphazardly collected images is that the variation in image quality, lighting, and posture among individuals of the same species can be quite large. In addition, the dataset consisted of images with complex backgrounds, which makes it difficult to distinguish the individual insects. This makes it necessary to use more complex and larger models such as Visual Geometry Group (VGG) [15], Inception [16], and Residual Networks (ResNet) [17] to perform classification reliably. Furthermore, training an effective model is challenging with rare species, where there is not enough data. In our work, we present a customized CNN model, which is able to classify moth species based on images with a controlled illumination, background, and camera setup.

Chang et al. [14] and Xia et al. [18] have proposed deep CNNs to make fine-grained classification of insects including butterflies and moths based on images from the Internet. In addition to the challenges listed above, such images could be a mix of pinned specimens and live individuals that have very different posture. To train a classifier to distinguish among species visiting a light trap, the individuals should ideally be photographed in their natural resting position.

Zhong et al. [19] developed a trap based on a Raspberry Pi 2 model B with the corresponding Raspberry camera module. An algorithm was proposed for the detection and classification of different orders of flying insects. To detect and count the number of individuals, a first version of the deep neural network “You Only Look Once” (YOLO) [20] was used. A SVM was then used to perform classification of insect order based on features. Using a combination of YOLO and SVM minimized the amount of training data required, but the classification was only done to the taxonomic resolution of insect orders with a counting and classification precision of 92.5% and 90.8%, respectively. The classifications made with SVM were based on manually defined features. The method is, therefore, difficult to scale up to individual species classification (e.g., of moth species) as selecting useful features would be challenging.

Roosjen et al. [21] present an automated monitoring system of fruit flies in crops for pest management. The system uses image-based object detection through deep learning to identify the spotted wing *Drosophila* (*Drosophila suzukii*). The trained ResNet [17] model was able to perform sex discrimination based on 4753 annotated flies.

Here, we present a novel, automatic light trap with a camera and several illuminations to attract and record insects including moth species without killing them. The computer vision system is based on a Raspberry Pi and a high-resolution web camera, which is able to capture detailed images of the individual insects.

Our novel image processing pipeline incorporates the temporal dimension of image sequences. The customized CNN model only needs few training samples with use of data augmentation. Compared to the results in [11,18,22], we present a fully functional image processing pipeline that includes tracking of insects and utilizes a CNN for moth spices classification in the intermediate steps of the pipeline. The novel image processing algorithm named Moth Classification and Counting (MCC) is able to track individuals and thus minimizes double counting.

## 2. Materials and Methods

The system was designed to attract moths and insects during the night and automatically capture images based on motion. Whenever a change within the camera field of view was detected by the computer vision system a sequence of images of the moth or insect was captured and stored on a hard drive. Any insect above a certain size would be detected by the camera and insect motion would be captured. The construction and selection of components of the system and algorithm to count and identify the species of moth are described in the following sections.

### 2.1. Hardware Solution

The primary components of the light trap vision system were the ultraviolet (UV light) fluorescent tube from Bioform [23] (Article No.: A32b), the light table from Computer-mester [24], and the computer vision system as illustrated in Figure 1. The purpose of the UV light was to attract insects to the location of the trap from a long distance. A light table (LED A3 format) placed in front of the computer vision system was covered with a white sheet to ensure a diffuse illumination without reflection in the images. The white uniform background ensured easy detection of the insects and the light table helped to ensure that the insects settled on the white sheet. The computer vision system was composed of a light ring from 24shop [25], web camera from Logitech [26] (Brio Ultra HD Pro Webcam) with a resolution of 3840 × 2160 pixels, and a Raspberry Pi 4 computer. The light ring (diameter 153/200 mm) was placed in front of the light table to ensure a diffuse foreground illumination of the insects. The intensity of the light ring could be adjusted from a combination of yellow and blue/white LED light. Camera focus and exposure was manually adjusted to achieve an optimal image quality as seen in Figure 2. The field of view was adjusted in order to ensure sufficient resolution of the insects and cover approximately half of the area of the light table. A working distance of 200 mm from camera to light table was chosen. This gave a field of view of 320 × 170 mm and resulted in a sufficient image quality for identifying moth species. The camera and light were adjusted so that the moth species could be identified by an expert based on an enlarged photo.

On some of the nights sugar water was sprinkled on the white sheet to attract insects. The sugar water caused more insects to stay on the sheet and for a longer period of time. However, the trap also worked without sugar water, although less effectively. A motion program [27] running on the Raspberry Pi 4 was installed to capture a sequence of images whenever a movement was detected in the camera view. It was programmed to save images in JPG format whenever it detected change in more than 1500 pixels. This setting ensured that only larger insects were captured and thus filtered smaller insects such as mosquitoes and flies from the footage. The maximum frame rate was limited to 0.5–2 fps, resulting in a time frame of 0.5–2 s between images. On warm summer nights with a high level of insect activity, more than 20,000 images were captured per night. To save space on the hard drive, a frame rate of 0.5 fps was selected, which was sufficient for identifying and counting moth species. The saved images from the hard drive were collected and processed offline on a remote computer by the MCC algorithm.

To save power, the Raspberry Pi was programmed to capture images between 9 p.m. and 5 a.m. and to turn off the UV light during daytime. A circuit with relay was constructed to control the UV light and placed inside the computer and camera box. A solid-state drive (500 GB) was connected to the Raspberry Pi to store the captured images. Inside the junction box, a DC-DC converter was installed to convert the supply voltage from 12 V to 5 V. The Raspberry Pi and light ring was supplied with 5 V and UV light, and the light table was supplied with 12 V.

The whole system was supplied with 12 V. The power consumption was 12.5 W during daytime when the UV light was turned off and the computer system was idle. During the night, the system used 30 W when the UV light was turned on and the computer vision system was recording images. With a power consumption of only 12.5 W during day hours and 30 W during night hours, a 12 V battery with a 55–100 Wp solar panel and regulator should be sufficient for powering the system during the entire summer period.

### 2.2. Counting and Classification of Moths

We developed a novel computer vision algorithm referred to as Moth Classification and Counting (MCC), which was able to count the number of insects and identify known species of moths on an offline remote computer. (Open source code: https://github.com/kimbjerge/MCC-trap).

However, we note that it was possible for individual moths to be counted multiple times if they left the camera field of view and then returned to the light table later. The algorithm produced data of individual moths and their species identity, as well as the number of unknown insects detected within a recording period. For every individual moth detected, the time and identity was recorded. In the following subsections, important parts of the algorithm are explained in more detail. However, we will first present a brief overview of the MCC algorithm. The MCC algorithm was composed by a number of sequential steps, where each image in the recording was analyzed as illustrated in Figure 3.

The first step was to read an image from the trap segmented as black and white, followed by blob detection to mark a bounding box around each detected insect as described in Section 2.2.1. The position of each insect region in the image was estimated based on the center of the bounding box.

The second step tracked multiple insects in the image sequence as described in Section 2.2.2. Tracking was important for recording the movement and behavior of the individual insect in the light trap and ensuring that it was only counted once during its stay in the camera’s field of view.

For every insect track, the algorithm evaluated whether the insect was a known moth species. A trained customized CNN, which used a fixed cropped and resized image of each insect, was used to predict moth species. This third step is described in more detail in Section 2.2.3.

The final step collected information and derived a summary of counted individuals of known moth species and unknown insects detected and tracked by the algorithm. The summary information was annotated to the image with highlighted marks of each insect tracks.

#### 2.2.1. Blob Detection and Segmentation

A grayscaled foreground image was made by subtracting a fixed background image of the white sheet without insects. The methods of using a global threshold or regions of adaptive threshold segmentation were investigated to perform segmentation of a black and white image. The Otsu [28] threshold algorithm turned out to be the best choice. A black and white image was made using Otsu threshold on the foreground image, followed by a morphological open and close operation to filter small noisy blobs and closing of blobs. Finally, the contour of blobs was found and the bounding box of insect regions was estimated. In rare cases, when two or more individuals were too close together, the estimation of the position based on bounding boxes was unsuccessful. A result of segmentation and bounding box estimation is shown in Figure 4.

#### 2.2.2. Insect Tracking and Counting

Tracking was used to reduce each visit of an insect in the camera field of view to one observation. However, we note that an individual insect could be counted again if it left the light trap and returned later during the night.

The individual insects were relatively stationary during their stay, and images were captured at two second intervals in case of activity. Therefore, it was assumed that a match was more likely for the shortest distance between two bounding boxes. That is, two boxes that were close to each other were likely to be the same individual.

The position and size of each insect were estimated for every single frame, and tracking could therefore be solved by finding the optimal assignment of insects in two consecutive images. The Hungarian Algorithm [29] was our chosen method for finding the optimal assignment for a given cost matrix. In this application, the cost matrix should represent how likely it was that an insect in the previous image had moved to a given position in the current image. The cost function was defined as a weighted cost of distance and area of matching bounding boxes in the previous and current image. The Euclidean distance between center position (x,y) in the two images was calculated as follows.
(1)$$Dist=\sqrt{{\left(x_{2}-x_{1}\right)}^{2}+{\left(y_{2}-y_{1}\right)}^{2}}$$

This distance was normalized according to the diagonal of the image:(2)$$Max_{dist}=\sqrt{{\left(I_{height}\right)}^{2}+{\left(I_{width}\right)}^{2}}$$

The area cost was defined as the cost between the size of bounding boxes:(3)$$Area_{cost}=\frac{Min_{area}}{Max_{area}}$$

A final cost function in Equation (Equation 4) was defined with a weighted cost of distance $W_{dist}$ and weighted cost of area $W_{area}$.
(4)$$Cost=\frac{Dist}{Max_{dist}}W_{dist}+\left(1-Area_{cost}\right)W_{area}$$

The Hungarian Algorithm required the cost matrix to be squared and, in our case, was defined as an N×N matrix, where each entry was the cost assigning $insect_{i}$ in previous image to $insect_{j}$ in current image. After a match with minimum cost, the entry in the current matrix was assigned a Track ID from the entry in the former. The found Track IDs and entries were stored and used in the upcoming iteration. Dummy rows and columns were added to the matrix to ensure that it was always squared. All entries in the dummy row or column had to have a cost significantly larger than all other entries in the cost matrix to ensure that the algorithm did not make a “wrong” assignment to a dummy. The insect assigned to a dummy could be used to determine which insect from the previous image had left, or which insect had entered into the current image.

To evaluate the performance of the tracking algorithm, two metrics were defined based on the work in [30]. The measure False Alarm Rate (*FAR*) was an expression of the probability that a given track was incorrect. It describes the number of false alarms relative to the total number of tracks, that is, how many times the tracker made a wrong track compared to the times it made a track.
(5)$$FAR=\frac{FP}{TP+FP}$$

While a True Positive (*TP*) was defined as an individual who retained its uniquely assigned Track ID in its entire presence of the observation, a False Positive (*FP*) was defined as an individual who was either counted multiple times or assigned a new Track ID.

The term Tracking Detection Rate (*TDR*) is a measure of the number of individuals who maintained their own Track ID in relation to the established Ground Truth (*GT*), during the course of observation. The size was therefore used as the primary scale to express the tracker’s ability to maintain the same Track ID for the individual insects in an observation.
(6)$$TDR=\frac{TP}{TG}$$

*GT* was defined as the total number of unique individuals in the test set measured by manual counting.

#### 2.2.3. Moth Species Classification

In the field of deep learning, specific architectures of CNNs have provided particularly positive results in many areas of computer vision [31]. CNNs use both pixel intensity values and spatial information about objects in the image. It was a challenging task to find a suitable CNN architecture for classification of moth species. Based on an investigation of several CNN architectures [32,33], a customized network was designed inspired by the work in [34]. Hyperparameters of the architecture were explored to find the optimal network architecture to classify moth species. The model was designed to be light and fast for the purpose of being able to be executed on the embedded Raspberry Pi computer used in the light trap.

As the camera recorded images of insects with a constant working distance, the insects did not change in size in the images. The moths were labeled with bounding boxes with an average size of 368 × 353 × 3 pixels and a standard deviation of 110 for pixel height and width. Initial experiments gave poor results with a resized input size of 224 × 224 × 3, which many CNNs [35] use. Improved results were achieved by reducing the input size, while still being able to visually identify the moth species. Based on the given camera setup the bounding boxes were finally resized approximately three times to a fixed window size of 128 × 128 × 3 as input for the customized CNN model.

Only 2000 images with a even number of images for eight different classes of moths were used for training the CNN model. A customized model was designed to work with this limited amount of training data.

The CNN model had four layers for feature detection and two fully connected layers for final species classification. The optimal architecture was found by using combinations of hyperparameters for the first and last layer in the CNN. Below are the parameters used to train different CNN’s for species classification.

Fixed pool size and stride, n×n, n∈{2}Kernel size n×n, n∈{1,3,5}Convolutional depth *n*, n∈{32,64,128}Fully connected size *n*, n∈{256,512}Optimizer *n*, n∈{Adam,SGD}

The optimal chosen CNN architecture is shown in Figure 5. The first layer performed convolution using 32 kernels with a kernel size of 5 × 5 followed by maximum pooling of size 2 × 2 and stride 2. All the following layers used a kernel size of 3 × 3. The second and third layer performed convolution using 64 kernels with the same pooling size as mentioned above. The final layer also used 64 kernels based on the optimization of hyperparameters. All convolutional layers used the Rectified Linear Unit (ReLu) activation function. The last fully connected layer had two hidden layer with 4096 and 512 neurons and a softmax activation function in the output layer. Two of the most commonly used optimizers—Adaptive Moment Estimation (Adam) and Stochastic Gradient Decent (SGD)—were investigated. While Adam was an optimizer that converged relatively quickly, it did so at the expense of a greater loss. SGD, on the other hand, converged more slowly, but achieved a smaller loss.

Developing a deep learning model for classifying species was an iterative process with an alternation between selecting images and training CNN models. When selecting and annotating images, experiments showed that it was important to vary images with different individuals and preferably equal numbers of each species. From the experiment mentioned in Section 2.3, images of nine different frequently occurring insect species were selected to train the different CNN models. The classes of species were chosen based on the recorded data from where a sufficient number of images could be selected to train the CNN models. According to our screening of the data, no other species were found in sufficient quantity to allow for their inclusion in the dataset. Examples of images of each individual species are shown in Figure 6.

The dataset was created by selecting images of different individual insects from the captured sequences of images in the period of observation. To collect sufficient number of samples for each class several images with different orientations of the same individual was chosen. With the use of sugar water, the dataset also contained many wasps (*Vespula vulgaris*). Therefore, they were included as a separate class. A separate class containing a variation of cropped background images was also included to classify false blob detections without insects. A total of ten classes was used for training with nine different insects including seven different moth species and one subtribe. These images were annotated with a bounding box for each individual, and the moth species determination was reviewed by an expert from the image only.

Table 1 shows an overview of the occurrence of all species in the chosen dataset for training and validation of the CNN algorithm. The moth class *Hoplodrina* complex consists of the three moth species *Hoplodrina ambigua*, *Hoplodrina blanda*, and *Hoplodrina octogenaria*. These species are very similar in appearance and it was too difficult to distinguish between them from the images alone.

It was a challenge to obtain a sufficient number of images. Especially, *Agrotis puta* and *Mythimna pallens* had fewer occurrences than the other species. That was the main reason for the limited number of images (250) for each species. Data augmentation was therefore applied to all images with a flip vertical, horizontal, zoom, different illumination intensity and rotation of different degrees. This operation provided more training data and was used to create a uniform distribution of species. The dataset was scaled with a factor of 32 times, resulting in 72,000 images, where each class contained 8000 data points after augmentation. From this dataset, 80% was used for training and 20% for validation of the CNN model.

To find the best CNN architecture for species classification, different hyperparameters were adjusted as described in Section 2.2.3. A total of 64 architectures were trained using a dropout probability of 0.3 after the second to last hidden layer. The average $F_{1}$-score for all classes was used as a measure for a given architecture’s performance.

The five best architectures had high $F_{1}$-scores, which only varied by 0.02, but had a varying number of learnable parameters (Table 2). Compared to SGD, Adam turned out to be the superior optimizer for training of all models. In the end, the architecture that had a rating among the three highest $F_{1}$-score but the lowest amount of learnable parameters (2,197,578) was chosen. The reason for this is that an architecture with many parameters and few training data would increase the risk of overfitting the neural network.

The chosen model shown in Figure 5 had an $F_{1}$-score of 92.75%, which indicated that the trained CNN was very accurate in its predictions. This final architecture was chosen because it achieved average precision, recall, and an $F_{1}$-score of 93%, which indicated a suitable model classification.

The confusion matrix (Figure 7) was based upon the validation of the chosen model. The confusion matrix has a diagonal trend, which indicates that the model matched the validation set well. The model had a recall of 93%, indicating that only 7% of the moth species in the validation set were missed. A similar precision of 93% was obtained, indicating that only 7% were wrongly classified.

Finally, the customized CNN architectures were compared with selected state-of-the-art CNN optimized architectures. EfficientNetB0 [36] is scaled to work with a small image input size of 224 × 224 pixel and has 4,030,358 learnable parameters. Using the moths dataset with the same data augmentation, the EfficientNetB0 achieved a $F_{1}$-score of 88.62%, which is lower than our top five best architectures. DenceNet121 [37] with 7,047,754 learnable parameters gave a $F_{1}$-score of 84.93% which is even lower. CNN architectures with many parameters (more than 20,000,000) such as ResNetV50 [38] and InceptionNetV3 [39] gave a high training accuracy, but a lower validation $F_{1}$-score of 69.1% and 81.7%, respectively. This result indicates overfitting and that more training data are needed when such large deep learning networks are used. A very high $F_{1}$-score of 96.6% was finally achieved by transfer learning on ResNetV50 using pretrained weights and only training the output layers. This indicates that the state-of-the-art was able to outperform our proposed model, but requires pretrained weights with many more parameters.

#### 2.2.4. Summary Statistics

The detection of insects was summarized based on the number of counted insects, the number of moth species found, and the number of unknown insects found (i.e., unknown to the trained CNN algorithm). The statistics were updated as the images were analyzed in order to enable visual inspection of the result during processing. Thus, the statistics were always updated throughout the execution of one iteration of the algorithm see Figure 3. The classification phase simultaneously classified each individual and assigned labels to each individual species. That is, an individual in one image could be classified as a different species than the same individual in the previous image. This phase ensured that the moth species most frequently classified in a track was represented in the final statistics. An insect was only counted if it was observed in more than three consecutive images thus ignoring noise created by insects flying close to the camera. Several other parameters were defined and adjusted to filter similar noisy tracks created by very small insects.

### 2.3. Experiment

An experiment was conducted in Aarhus, Denmark, in the period 3 August to 23 September 2019, capturing 272,683 images in total over 48 nights. The constructed light trap was located close to the forest of Risskov, near Aarhus, Denmark, where it was activated in the period from 9 p.m. to 5 a.m. each night. The collection of image data was limited to a smaller sample of the family Noctuidae moths occurring in Denmark. A survey was defined as the images from a single night’s footage. On average, one survey consisted of 5675 images. It should be noted that the number of images captured in the surveys varied significantly. The smallest survey consisted of 134 images, while the largest consisted of 27,300.

All moth detections made by the trap during the period 19 August to 23 September were summarized in daily counts as a measure of their abundance. The first two weeks (3–17 August) are not included in this summary, as this period was used to perform system test, camera and light adjustments. In addition to describing variation in abundance of each of the species, we used these data to analyze how sensitive each species was to variation in weather conditions. Specifically, hourly data on air temperature, relative humidity, and wind speed from the weather station of the Danish Meteorological Institute in Aarhus was summarized for the exact hours (9 p.m. to 5 a.m.) during each night when the moth recordings were taking place. These weather variables have previously been identified as important for the flight activity of moths [9]. The three weather variables were used as predictors in generalized linear models of log-transformed daily values of each of the eight moth species detected by the trapping system. All statistical analyses were performed in the R 4.0.2 platform [40].

## 3. Results

In the following sections, the results from the experiment, and the last stages of the algorithm concerning tracking, statistics (as described in Figure 3), and temporal variation in moth abundance will be presented.

### 3.1. Insect Tracking and Counting

Tracking was tested on a survey consisting of 6000 different images distributed over 122 min of insect activity. Images from this survey was not used for training and validation of the CNN model. This survey was collected on the night between 25 August and 26 August 2019, and represents the average survey for a single night. The set was selected in order to provide the tracking algorithm with a sufficient challenge. The survey was challenging because the individual insects moved in and out of the camera view repeatedly. In addition, there were examples of two insects siting close together. Furthermore, some insects flew very close to the camera lens. The algorithm was therefore challenged in its ability to retain the correct tracking. The survey was manually reviewed to establish a Ground Truth (GT) thereby enabling evaluation of the performance. In total, a GT of 82 individual moths was observed.

The tracker algorithm measured 83 individual tracks, where 65 were identified as True Positive (TP) and 18 as False Positive (FP) moth tracks. Table 3 shows a calculated Tracker Detection Rate (TDR) of 79% and False Alarm Rate (FAR) of 22% based on the survey of 6000 images.

### 3.2. Summary Statistics

To evaluate the final system, including tracking and classification, three survey nights were selected (30 August to 2 September 2019). This period contained a high number of moths and a high variation of species, wherein all eight moths species were represented. The duration of the survey every night was from 22:30 to 05:00, adding up to a total of 22.5 h. Fewer than 1% of the images from this period of survey were used to train the CNN model. In this way, the algorithm was tested on a mostly unknown data material.

These surveys represent a typically night with high activity, where 11,263, 8195, and 10,767 images were collected in the trap. The images were studied to establish a GT, which was obtained by reviewing the same images that the algorithm analyzed. This involved a manual count of occurrences for each moth species and comparing it to the MCC algorithm result. A video for each night was created with time stamp, Track ID, bounding boxes, and annotation of predicted species. These videos were studied manually for each Track ID to evaluate the prediction against a visual species classification. Table 4 shows the algorithm’s estimate of the number of moth species and the established GT of the survey.

There were also unknown moth species and other insects in the dataset, especially mayflies and harvestmen. However, only few of them were detected correctly. Table 5 shows the algorithm’s precision, recall, and $F_{1}$-score based on the tracked and predicted moths in Table 4. In total, a precision of 69%, a recall of 74%, and an $F_{1}$-score of 71% were achieved.

### 3.3. Temporal Variation in Moth Abundance

The seasonal dynamics of each of the eight species of moths detected by the MCC algorithm showed clear similarities with peak abundance during the last days of August 2019 (Figure 8). However, *Autographa gamma* observations were restricted to mostly a single night, while other species were frequent during several consecutive nights. Some species including the *Hoplodrina* complex exhibited a second smaller peak in abundance around 10 September 2019. This variation was mirrored to some extent in the weather patterns from the same period (Figure 9). The statistical models of the relationships with weather patterns revealed that abundance of all species was positively related to night temperatures, while three species also responded to wind speed, and another three species responded to both air humidity and wind speed (Table 6).

## 4. Discussion

In this study, we have presented an Automated Moth Trap (AMT) for efficient monitoring of nocturnal insects attracted to light including many moth species. The AMT is widely applicable as a means of generating standardized insect monitoring data. Furthermore, we have shown that it is capable of generating novel insights into the seasonal dynamics and weather sensitivity of individual moth species. Further developments can expand the discriminatory abilities of the CNN classification network, but we note that the current solution can already capture the relevant image data, while the deep learning models can be refined further.

The tracker measured a Tracking Detection Rate (TDR) of 79%, which means that it tracked the majority of the observed insects correctly. However, the algorithm had a False Alarm Rate (FAR) of 22%, which means that nearly one-quarter of the detected tracks were incorrect. This test was based on one night with many challenges and variations of movements.

Over three nights with 485 counted moths, the MCC algorithm measured an average combination of tracking and classification $F_{1}$-score of 0.71, where recall was 5% higher than precision. This score of the whole system could be improved as especially many unknown insects where classified wrongly. A recall of 0.16 for unknowns as seen in Table 5, meaning that 84% of the unknown insects were not detected. Table 4 shows that *Amphipyra pyramidea* has a particularly large number (50) of False Positives (FPs) and a low precision of 0.18. Many of these classifications were really *Noctua pronuba*, *Xestia c-nigrum*, or unknowns, which explains the lager number (25, 28, and 48) of False Negative (FN) for these three classes. All in all there were many false classifications, especially in cases where there were variations in the color, visibility or orientation of a species.

One source of error was related to wrong blob detection and segmentation resulting in many of the false classifications. Another frequent source of error was individual insects moving at the edge of the camera’s field of view. This resulted in insects disappearing completely or partially from an observation only to appear again later. An improvement could be to ignore insects that are partly visible at the edges of the image. Furthermore, errors occurred in cases where an insect flew close to the camera lens. In such cases, the algorithm could place multiple boxes on a single individual and make a match with these fake boxes. However, because the insects flying in front of the lens rarely stayed in the field of view for more than a few frames, the design of the tracker often prevented this type of error. An individual insect had to be detected in at least three images before it was counted. Consequently, a flying insect appearing in only a few frames was below the threshold filter value, and the final statistics were not affected. The last identified source of error occurred when two individuals were located closely together in the image. In this case, the tracking algorithm could not separate the two individuals and therefore only placed one bounding box.

One of the potential sources of error in the algorithm was the choice of dataset used for training and validation of the CNN model. Collecting a sufficiently large dataset with enough data points for efficient classification of the rarer species was a significant challenge. However, it is likely that increasing the number of images available for training the CNN model could improve performance even further. The current classification algorithm relies heavily on padding the bounding box found during blob segmentation. The performance of the system changes significantly with variation in padding before CNN prediction. The CNN algorithm was trained on a dataset using manual annotations of the moths. These do not surround the moths as closely as the bounding boxes placed by the blob segmentation (see Figure 4 and Figure 6). Thus, there is a difference in the sizes.

Our moth trap was able to detect distinct night-to-night variation in abundance across the 36 day monitoring period during August–September 2019 for the seven species and one species complex of moths. The abundance variation was also significantly related to weather variables measured during the hours when the trap was active. Our results support previous observations of positive effects of temperature, generally negative effects of wind speed, and negative effects of air humidity in a few species [9]. As this system can be implemented across full seasons, the high temporal resolution of the data collection will enable fine-scale analyses of species-specific sensitivity to weather variables as well as detailed phenological variation and interannual variation in abundance [41].

The proposed AMT and MCC algorithm is promising as an automated system for observations of nocturnal insects. The system could be improved by creating more training and validation data for the CNN classification algorithm, but the system should already be useful for ecological studies where traps are placed at sites with contrasting ecological conditions and habitats.

## 5. Conclusions

In this paper, we have presented a light trap and computer vision system for monitoring live insects and moth species. The automated light trap was composed of a web camera, Raspberry Pi computer, and special illumination to attract insects during the night. The camera and illumination were adjusted to achieve a high resolution and quality of images of eight different moth species. The light trap recorded more than 250,000 images during 48 night over the summer 2019 with an average of 5675 images per night.

A customized CNN was proposed and trained on 2000 labeled images of live moths. This allowed us to detect and classify eight different moth species. The network achieved a high validation $F_{1}$-score of 0.93. The algorithm measured an average classification and tracking $F_{1}$-score of 0.71 and a tracking detection rate of 0.79. This result was based on an estimate of 485 moths observed in 30,225 images during three nights with 22.5 h of insect activity.

Furthermore, the paper identified potential improvements to the system such as the ability to handle partly visible insects. The amount of training data for the presented CNN model for species classification was highlighted as a focus area.

Overall, the proposed light trap and computer vision system showed promising results as a low-cost solution, which offers non-destructive and automatic monitoring of moths and classification of species. As such, the system provides novel opportunity for image-based insect monitoring of global relevance [42]. It should be considered as a viable alternative to traditional methods which typically requires tedious manual labor (i.e., visiting the trap several times in a season for observation) and often results in the killing of rare species of insects.

## Figures and Tables

**Figure 1 sensors-21-00343-f001:**
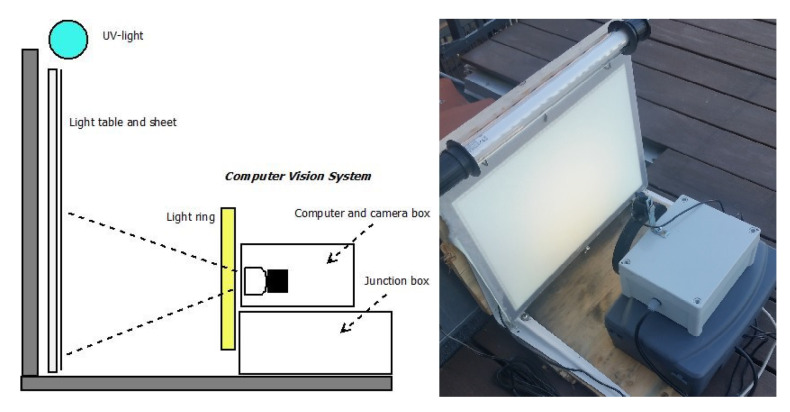
The portable light trap with a light table, a white sheet, and UV light to attract live moths during night hours. The computer vision system consisted of a light ring, a camera with a computer and electronics, and a powered junction box with DC-DC converter.

**Figure 2 sensors-21-00343-f002:**
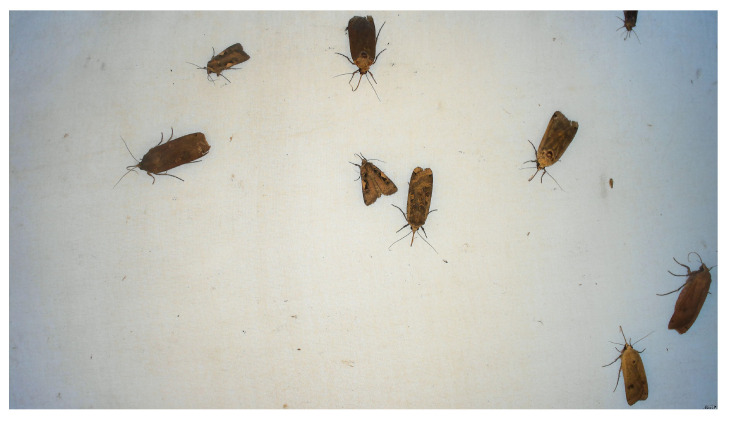
A picture of 3840 × 2160 pixels taken by the light trap of the light table and nine resting moths.

**Figure 3 sensors-21-00343-f003:**
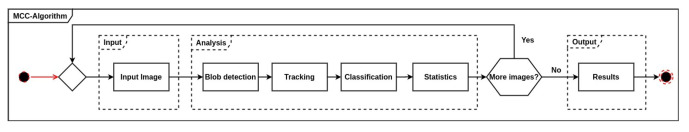
The processing pipeline of the intermediate- to high-level image processing algorithms to track and count the number of insects. A customized trained Convolutional Neural Network (CNN) was used for the moth species classification.

**Figure 4 sensors-21-00343-f004:**
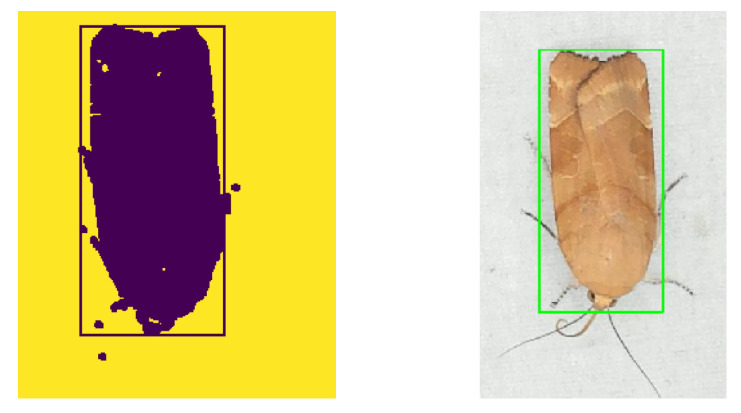
A blob segmented image of a detected moth (*Noctua fimbriata*) with bounding box.

**Figure 5 sensors-21-00343-f005:**
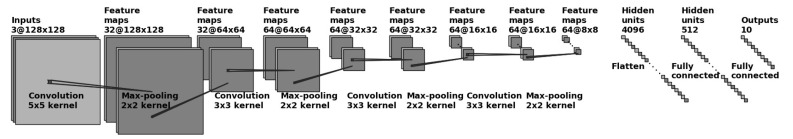
CNN architecture for moth species classification used an input of a 128 × 128 RGB image.

**Figure 6 sensors-21-00343-f006:**
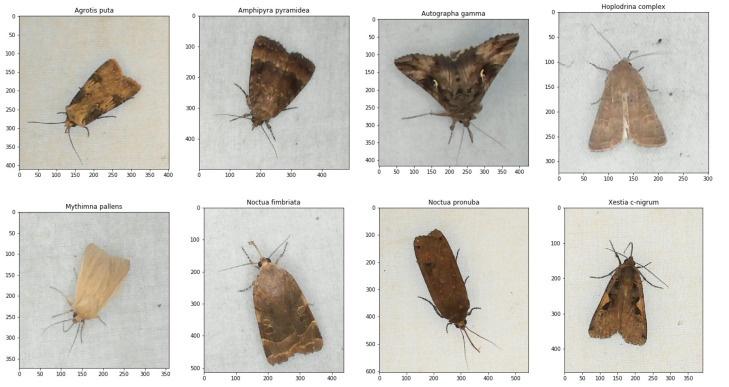
Examples of the eight moth species that were observed and labeled for training of the CNN models. The axes indicate the number of pixels of the annotated bounding boxes.

**Figure 7 sensors-21-00343-f007:**
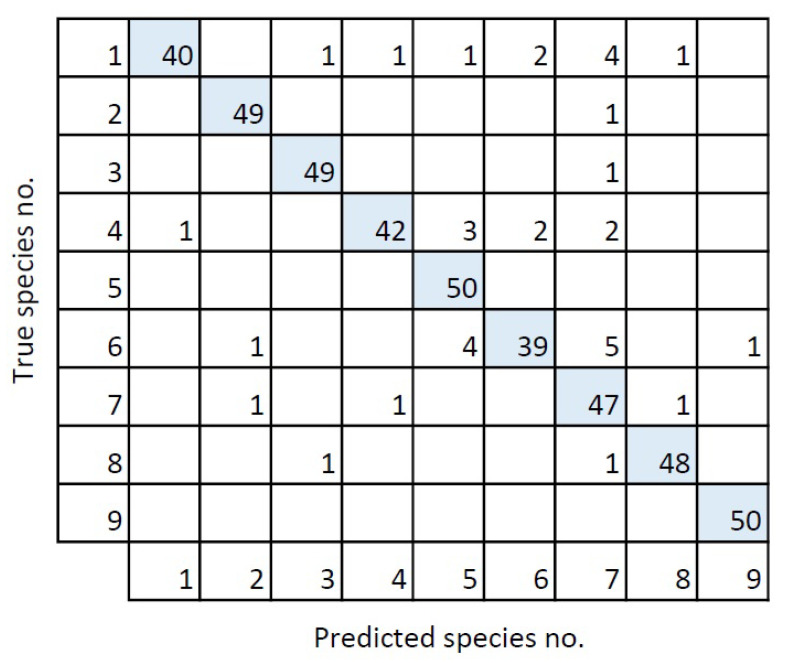
Confusion matrix for the validation of the best model. The numbers corresponds to the species numbers in Table 1.

**Figure 8 sensors-21-00343-f008:**
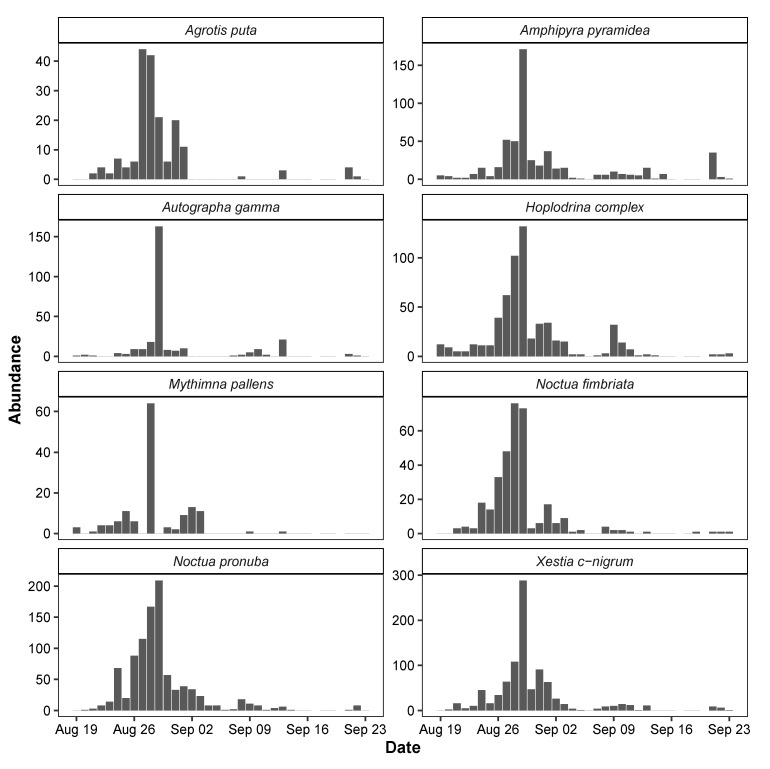
Observation of moths during night (9 pm–5 am) in Aarhus, Denmark, in the period from 19 August to 23 September 2019.

**Figure 9 sensors-21-00343-f009:**
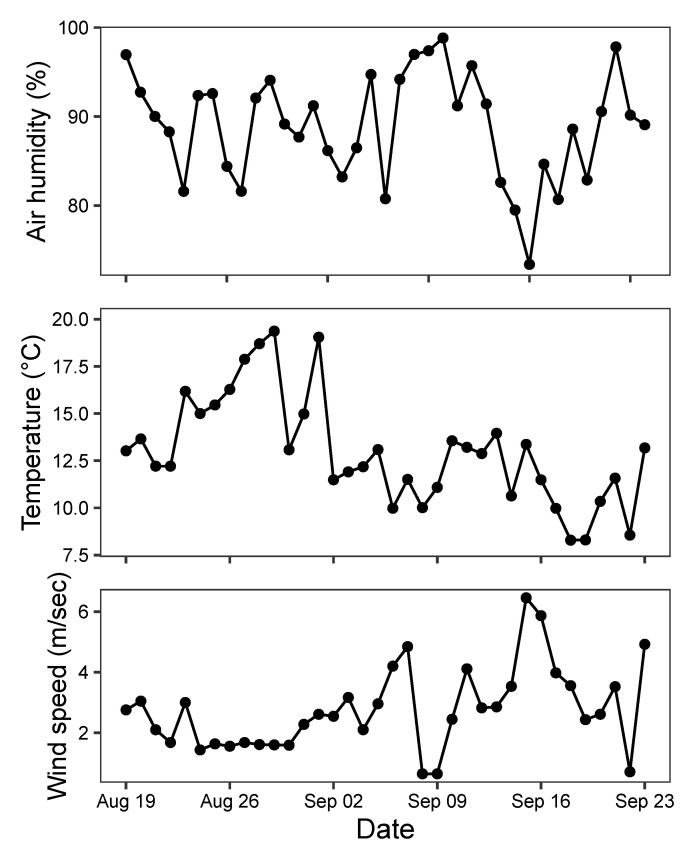
Weather conditions during night (9 p.m.–5 a.m.) in Aarhus, Denmark, in the period of observation from 19 August to 23 September 2019.

**Table 1 sensors-21-00343-t001:** Number of insect species in the dataset used for training and validation.

No.	Species	Numbers
1	*Agrotis puta*	250
2	*Amphipyra pyramidea*	250
3	*Autographa gamma*	250
4	Hoplodrina complex	250
5	*Mythimna pallens*	250
6	*Noctua fimbriata*	250
7	*Noctua pronuba*	250
8	*Xestia c-nigrum*	250
9	*Vespula vulgaris*	250
	**Total**	2250

**Table 2 sensors-21-00343-t002:** Ranking of the CNN architectures with highest and lowest $F_{1}$ classification scores. Rank 1 to 32 were trained using the Adam optimizer. Rank 33 to 64 were trained using the SGD optimizer. The hyperparameters column shows values of {kernel size layer 1, kernel size last layer, convolutional depth layer 1, convolutional depth last layer, fully connected size}.

Rating	Hyperparameters	Learnable Parameters	$F_{1}/$-Score
1.	3, 3, 32, 128, 512	4,330,122	0.942
2.	5, 1, 32, 128, 512	4,266,122	0.929
3.	5, 3, 32, 64, 512	2,197,578	0.928
4.	3, 3, 32, 64, 512	2,196,042	0.923
5.	5, 3, 32, 128, 512	4,331,658	0.922
...			
31.	5, 1, 64, 64, 512	2,185,674	0.871
32.	5, 3, 32, 32, 512	2,164,810	0.871
33.	5, 3, 64, 128, 512	4,352,522	0.853
34.	5, 3, 32, 128, 512	4,331,658	0.853
...			
62.	5, 1, 64, 64, 512	2,185,674	0.749
63.	3, 3, 64, 64, 256	1,163,978	0.737
64.	3, 3, 64, 64, 512	2,215,370	0.682

**Table 3 sensors-21-00343-t003:** Result of the tracking algorithm. From a Ground Truth (GT) of 82 moth tracks, 65 were identified as True Positive (TP) and 18 as False Positive (FP). The Tracker Detection Rate (TDR) and False Alarm Rate (FAR) is calculated.

Metric	Equation	Result
**TDR**	(Equation 6)	0.79
**FAR**	(Equation 5)	0.22

**Table 4 sensors-21-00343-t004:** Number of individuals of each moth species in the test dataset detected by the MCC algorithm and Ground Truth (GT). A True Positive (TP) is defined as an insect that retained the same Track ID and correct classification in the entire observation. A False Positive (FP) is defined as an insect that was incorrect classified or counted several times. A False Negative (FN) is defined as an insect that was not detected and classified correctly in the final statistic.

Moth Species	GT	TP	FP	FN
*Agrotis puta*	32	24	6	8
*Amphipyra pyramidea*	13	11	50	2
*Autographa gamma*	6	5	9	1
Hoplodrina complex	77	63	14	14
*Mythimna pallens*	5	5	10	0
*Noctua fimbriata*	18	16	9	2
*Noctua pronuba*	93	68	32	25
*Xestia c-nigrum*	184	156	18	28
Unknown	57	9	10	48
**Total**	485	357	168	128

**Table 5 sensors-21-00343-t005:** Precision, recall, and $F_{1}$-score based on the test dataset of three nights survey consisting of 30,225 images.

Moth Species	Precision	Recall	$F_{1}$-score
*Agrotis puta*	0.80	0.75	0.77
*Amphipyra pyramidea*	0.18	0.85	0.30
*Autographa gamma*	0.36	0.83	0.50
Hoplodrina complex	0.82	0.82	0.82
*Mythimna pallens*	0.33	1.00	0.50
*Noctua fimbriata*	0.65	0.89	0.74
*Noctua pronuba*	0.68	0.73	0.70
*Xestia c-nigrum*	0.90	0.85	0.87
Unknown	0.47	0.16	0.24
**Total**	0.69	0.74	0.71

**Table 6 sensors-21-00343-t006:** Estimated significance for humidity, temperature, and wind speed on abundance of moth species. Figures marked with bold has a significant positive or negative impact.

Moth Species	Intercept	Humidity	Temperature	Wind
*Agrotis puta*	3.44	−0.056±0.02	0.279±0.04	−0.430±0.10
*Amphipyra pyramidea*	−3.48	0.022±0.03	0.306±0.05	−0.180±0.12
*Autographa gamma*	−4.74	0.034±0.03	0.258±0.05	−0.215±0.12
Hoplodrina complex	−0.89	−0.003±0.02	0.333±0.04	−0.454±0.10
*Mythimna pallens*	3.01	−0.039±0.03	0.158±0.06	−0.301±0.14
*Noctua fimbriata*	4.04	−0.057±0.02	0.300±0.04	−0.553±0.09
*Noctua pronuba*	5.17	−0.057±0.02	0.327±0.04	−0.813±0.11
*Xestia c-nigrum*	0.46	−0.015±0.03	0.346±0.05	−0.570±0.13

## Data Availability

The train and validation data is available online at https://github.com/kimbjerge/MCC-trap.

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
