# Peer review of "An Automated Light Trap to Monitor Moths (Lepidoptera) Using Computer Vision-Based Tracking and Deep Learning"

_sensors, 2021, doi:10.3390/s21020343_

Round 1

Reviewer 1 Report

This paper presents a new sensor for detecting insects (lepidopteras). The sensor is composed of a light panel with sugar to trap the individuals, a camera and a raspeberry to process the images from the camera.

The sensor is able to count the number of individuals (from 9 different species) during the twilight and night. It is a good sensor for identify and track these kind of insects, but it could be used with different ones with less effort.

Author Response

Thank you for this review and feedback.

Reviewer 2 Report

The selection of eight species of Noctuidae appears to be a bit arbitrary. It needs to be explained better why this selection was made. It remains unclear whether this is in any way representative for a broader range of moth species or whether it could be some sort of "cherry-picking" the most suitable data from a large data set.

The eight Noctuidae "species" chosen are in fact seven species and one subtribe (Caradrinina) that includes a complex of 3 similar species. However, it is often stated that "eight different species" were examined, and this is technically not correct: it is about 7 species and one species complex. Since all species in this complex belong to the genus Hoplodrina, it would make more sense to name this complex e.g. "Hoplodrina spp complex".

Formatting: All genus and species names should consistently be written in italics, but not higher level taxa such as families (as Noctuidae) and not subtribes (Caradrinia). In all cases, such formatting errors occurred.

The article is generally a bit lengthy. Parts of the introduction have almost review character, and it should be condensed to the most important findings.

The use of sugar water "caused more insects to stay on the sheet". I understand this point, but from a monitoring point of view this means that the sheet needs to be replaced every day and this appears to be not very practical. And since so many photos were made, would it indeed make a measureable difference whether a moth individual stays for a minute or half an hour?

Small points

  • Line 2: involve (not involves)
  • Line 25 ff: There are around 160.000 described species, and around 18500 (or 20.000) are butterflies. This results in a percentage of around 12% (not 5%)
  • Line 28: One of the most famous insect decline studies was carried out in Germany and is not cited (Hallmann et al.)
  • Line 40: This sentence "promises" that computer vision paper follow, but then the study by Jonason et al. is cited that is only methodologically interesting
  • Line 288 spelling errror Hoplodrine
  • Line 343: I would avoid a term such as "Danish moths", better: Noctuidae moths occurring in Denmark.

Author Response

We would like to thank you for this review with very relevant inputs and feedback, which we have incorporated in the new revision of the paper. See below detailed response to each topic raised by the reviewer.

------------------------------------

The selection of eight species of Noctuidae appears to be a bit arbitrary. It needs to be explained better why this selection was made. It remains unclear whether this is in any way representative for a broader range of moth species or whether it could be some sort of "cherry-picking" the most suitable data from a large data set.

Response:

The classes of species were chosen based on the recorded data from where a sufficient number of images could be selected to train the CNN models. According to our screening of the data, no other species were found in sufficient quantity to allow for their inclusion in the dataset”. These sentences has been added to section “2.2.3. Moth species classification”..

------------------------------------

The eight Noctuidae "species" chosen are in fact seven species and one subtribe (Caradrinina) that includes a complex of 3 similar species. However, it is often stated that "eight different species" were examined, and this is technically not correct: it is about 7 species and one species complex. Since all species in this complex belong to the genus Hoplodrina, it would make more sense to name this complex e.g. "Hoplodrina spp complex".

Response:

The subtribe “Caradrinina” is now change to “Hoplodrina complex” as suggested. Species replaced with classes in abstract and for training of the CNN model.

------------------------------------

Formatting: All genus and species names should consistently be written in italics, but not higher level taxa such as families (as Noctuidae) and not subtribes (Caradrinia). In all cases, such formatting errors occurred.

Response:

Only species names are now written in italics, families and subtribes are written in normal font.

------------------------------------

The article is generally a bit lengthy. Parts of the introduction have almost review character, and it should be condensed to the most important findings.

Response:

Introduction has been shortened substantially and the paragraph summarizing the paper by Jonason et al. has been removed.

------------------------------------

The use of sugar water "caused more insects to stay on the sheet". I understand this point, but from a monitoring point of view this means that the sheet needs to be replaced every day and this appears to be not very practical. And since so many photos were made, would it indeed make a measureable difference whether a moth individual stays for a minute or half an hour?

Response:

“However, the trap also worked without sugar water, although less effectively.” We have mentioned that the trap also worked without sugar water. We do not have measures on the impact for use of sugar water. The sheet is not replaced, but only sprinkled occasionally with sugar water. Whether it would make a measurable difference is left for future studies.

Text changed to: “Sugar water was some of the nights sprinkled on the white sheet to attract insects. The sugar water caused more insects to stay on the sheet and for a longer period of time.” Although it would make no difference, whether the individual stays from a minute or half an hour.

------------------------------------

Small points

  • Line 2: involve (not involves) (done)
  • Line 25 ff: There are around 160.000 described species, and around 18500 (or 20.000) are butterflies. This results in a percentage of around 12% (not 5%) (done)
  • Line 28: One of the most famous insect decline studies was carried out in Germany and is not cited (Hallmann et al.) (done)
  • Line 40: This sentence "promises" that computer vision paper follow, but then the study by Jonason et al. is cited that is only methodologically interesting (Summary of study by Jonason et al. removed)
  • Line 288 spelling errror Hoplodrine (done)
  • Line 343: I would avoid a term such as "Danish moths", better: Noctuidae moths occurring in Denmark. (done)

Response:

The above points are corrected in the new revision.

Reviewer 3 Report

Authors present an automatic moth trap using a CNN-based detection and classification system.

It is very interesting and the results are very well detailed. In general, the paper is very complete and I think it may be accepted after some minor revisions.

Only one important thing that the authors may include in order to justify the usefulness of the automatic trap: they may do a latency test that present the time elapsed by the classifier, so that the system can classify in real time and activate the trap before the moths go.

Author Response

Thank you for this review and feedback.

All images were stored on a hard drive in the automatic trap captured by the motion program as described in section “2.1 Hardware solution”. That means any movements in the camera field of view were recorded with a framerate of 0.5-2 fps on the hard drive if any insect was detected. We have added a sentence to make this approach more clear: “The saved images from the hard drive were collected and processed offline on a remote computer by the MCC algorithm.

The automatic trap does not require the MCC algorithm to process the images in real-time. Although it would be possible to run the MCC algorithm written in Python on the Raspberry Pi 4. The time elapse by the classifier is only 40 ms but the blob detection and adaptive segmentation part of the algorithm is very time consuming with a time elapse of 77 sec. per. image. Using the simpler global threshold segmentation method reduces the total MCC algorithm processing time to 2.7 sec. per image. If an average of 5,675 images were recorded per night, the MCC algorithm would then take 4.3 hours to process one nights of insect activity on the Raspberry Pi 4.  We have decided not to include these considerations in the paper since it is not relevant for the results and use of the Automated Moth Trap (AMT).